# Glycation of Plant Proteins: Regulatory Roles and Interplay with Sugar Signalling?

**DOI:** 10.3390/ijms20092366

**Published:** 2019-05-13

**Authors:** Julia Shumilina, Alena Kusnetsova, Alexander Tsarev, Henry C. Janse van Rensburg, Sergei Medvedev, Vadim Demidchik, Wim Van den Ende, Andrej Frolov

**Affiliations:** 1Department of Biochemistry, St. Petersburg State University, Saint Petersburg 199034, Russia; schumilina.u@yandex.ru (J.S.); alena_kyy@mail.ru (A.K.); alexandretsarev@gmail.com (A.T.); 2Department of Biotechnology, St. Petersburg Chemical Pharmaceutical University, Saint Petersburg 197022, Russia; 3Department of Bioorganic Chemistry, Leibniz Institute of Plant Biochemistry, 06120 Halle, Germany; 4Laboratory of Molecular Plant Biology, KU Leuven, 3001 Leuven, Belgium; henry.jansevanrensburg@kuleuven.be (H.C.J.v.R.); wim.vandenende@kuleuven.be (W.V.d.E.); 5Department of Plant Physiology and Biochemistry, St. Petersburg State University, Saint Petersburg 199034, Russia; ssmedvedev@mail.ru; 6Department of Plant Cell Biology and Bioengineering, Belarusian State University, 220030 Minsk, Belarus; dzemidchyk@bsu.by; 7Department of Horticulture, Foshan University, Foshan 528231, China

**Keywords:** advanced glycation end products (AGEs), thermal processing of foods, methylglyoxal, plant glycation, protein degradation, protein glycation, sugar signalling

## Abstract

Glycation can be defined as an array of non-enzymatic post-translational modifications of proteins formed by their interaction with reducing carbohydrates and carbonyl products of their degradation. Initial steps of this process rely on reducing sugars and result in the formation of early glycation products—Amadori and Heyns compounds via Schiff base intermediates, whereas their oxidative degradation or reactions of proteins with α-dicarbonyl compounds yield a heterogeneous group of advanced glycation end products (AGEs). These compounds accompany thermal processing of protein-containing foods and are known to impact on ageing, pathogenesis of diabetes mellitus and Alzheimer’s disease in mammals. Surprisingly, despite high tissue carbohydrate contents, glycation of plant proteins was addressed only recently and its physiological role in plants is still not understood. Therefore, here we summarize and critically discuss the first steps done in the field of plant protein glycation during the last decade. We consider the main features of plant glycated proteome and discuss them in the context of characteristic metabolic background. Further, we address the possible role of protein glycation in plants and consider its probable contribution to protein degradation, methylglyoxal and sugar signalling, as well as interplay with antioxidant defense.

## 1. Introduction

Historically, protein glycation is usually referred to as a non-enzymatic reaction of protein lysyl and arginyl residues with reducing carbohydrates and carbonyl products of their oxidative and non-oxidative degradation [1]. The glycation process, initially believed to be acting in a rather random and non-specific way, usually involves lysine and arginine [2,3] and, to a lesser extent, cysteine and tryptophan residues [4]. These reactions, also known as Maillard reaction of proteins, are strongly enhanced by temperature increases [5]. Thereby, proteins are involved in a complex network of reactions, among which the pathways of early and advanced glycation can be distinguished [6]. The term “early glycation” assumes reversible interaction of reducing sugars, aldoses and ketoses, with N-terminal and side-chain protein amino groups [7]. The resulting aldoamine and ketoamine intermediates easily lose water, forming more stable aldimines and ketoimines (Schiff bases) which undergo the so-called Amadori rearrangement, i.e., transfer of one proton from C1 to C2 yielding N-substituted 1-amino-deoxy-ketoses, the Amadori products (Figure 1) [7]. Glucose-derived Amadori compounds—amino-deoxyfructos-1-yl adducts, represent the best characterized glycation products [8]. Ketoamine intermediates undergo a similar reaction cascade, known as Heyns rearrangement [9], i.e., migration of a proton from C2 to C1 leading to the formation of 2-amino-deoxyaldos-1-yl adducts (Heyns products) [10,11]. These first relatively stable intermediates of glycation, known as “early glycation products” [12] are readily involved in further degradation, oxidation and cross-linking reactions, yielding “advanced glycation end products” (AGEs) [13]. AGEs are involved in pathogenesis of diabetes mellitus [14], Alzheimer’s disease [15], ageing and are also involved in thermal processing of foods [16]. In humans, accumulation of AGEs results in the development of systemic inflammation [17], atherosclerosis [18] and accompanies the pathogenesis of such diabetes complications as uremia [19], neuropathy [20], retinopathy and angiopathy [21]. Dietary AGEs, consumed with thermally processed foods and absorbed in blood, contribute to the systemic inflammatory response [22].

Oxidative degradation of Amadori and Heyns compounds, known as glycoxidation [23], represents one of the major AGE formation pathways which contributes essentially in advanced glycation in blood [24] and Maillard reaction during thermal processing of foods [25]. Degradation of early glycation products also yield α-dicarbonyls [26], in particular, methylglyoxal (MGO) [27], glyoxal (GO) [28] and 3-deoxyglucosone (3-DG) [29] (Figure 1). These highly-reactive compounds readily react with lysine, arginine and cysteine residues, directly yielding AGEs with much higher rates compared to reducing sugars. Besides the glycoxidative pathway, α-dicarbonyls can be formed by metal-catalyzed oxidative degradation of sugars (monosaccharide autoxidation) [27], Schiff bases (so-called Namiki pathway) [30] and via non-oxidative pathway, i.e., 1,2-enolization, dehydration and formation of 3-DG (Figure 1) [31].

Other important sources of dicarbonyls are the reactions of lipid and fatty acid peroxidation [32], metabolism of serine [33] and non-enzymatic conversion of glyceraldehyde-3-phosphate and dihydroxyacetone phosphate into MGO [34]. Due to the existence of multiple pathways of advanced glycation (Figure 1), AGEs demonstrate a high degree of heterogeneity and are represented with a variety of aliphatic, aromatic and heterocyclic structures, many of which have a cross-linking nature (Figure 2). Thereby, classification of AGEs is often based on their carbonyl precursors and/or intermediates. It is important to note that despite the term “end-products”, many AGEs are relatively unstable and are often featured with compromised detectability by most analytical techniques.

The deleterious physiological effects of AGEs can be mediated by two principal mechanisms including the formation of intra- and inter-molecular cross-links [35] and induction of multi-ligand pattern-recognition receptors [36]. Indeed, on the one hand, excessive glycation leads to dramatic changes in protein structure, resulting in enhanced rigidness and higher resistance to degradation that, in turn, leads to accumulation of cross-linked products in cells, intracellular matrix and body fluids [37]. On the other hand, AGE-modified proteins can serve as effectors of intracellular signalling pathways [38]. During recent decades, multiple membrane and soluble proteins were annotated as receptors for glycation products in mammals [39]. Receptors to advanced glycation end products (RAGEs), belonging to the sub-family of immunoglobulin-like receptors, represent the best characterized group of such molecules [40]. There are several types of receptors for AGEs: AGE-R complex, SR-A I/II, CD36 and others [39]. Upon interaction with receptors, AGEs trigger inflammatory response by activation of mitogen-activated protein kinase (MAPK-), janus kinases (JAC-) and mitogen-activated protein kinases/extracellular signal-regulated kinases MAPK/ERK -signalling pathways [41]. Complications of diabetes mellitus, especially nephropathies [42], could be successfully attenuated by application of AGE antagonists interacting with RAGEs [43,44].

Despite significant progress in the evaluation of formation pathways, structural analysis and physiological effects of dietary and clinically relevant AGEs in humans, animals and yeast [45], for plants, this information is still mostly unknown. This is quite surprising because plants, as autotrophic organisms, continuously produce sugars that may reach very high levels in plant cells at particular locations and under specific circumstances [46], probably necessitating the need for local detoxification and scavenging mechanisms as this is, for instance, the case in bacterial and Archaean (hyper)thermophiles [47]. Therefore, in this review, we summarize and interpret the first steps done in the field of plant protein glycation during the last decade. We also consider the work accomplished on dietary AGEs, discovered in plant-derived foods, as an important step in understanding of structural, analytical and nutritional aspects of glycation in the Plant Kingdom. Finally, we discuss possible physiological roles of plant protein glycation in the context of future perspectives and new directions in the field of plant glycation research.

## 2. AGEs in Plant-Derived Foods

The demand for high quality of daily consumed foods is continuously growing along with general improvement of quality of life. In this context, not only appearance and taste of meals, however also their quality and safety become important for the consumer’s choice. On the other hand, thermal processing (i.e., heat treatment) of conventional foods represents an integral part of their preparation. Due to the pronounced temperature dependence of the Maillard reaction rate [26], this procedure results in dramatic increase of AGE contents in thermally processed foods [48].

In general, *N^ε^*-(carboxymethyl)lysine (CML) is recognized as the main marker of AGE formation in foods [49]. Typically, quantitative determination of this compound relies on enzyme-linked immunoassay (ELISA) [50] or liquid chromatography, coupled on-line to tandem mass spectrometry (LC-MS/MS) [51,52]. For example, implementation of the LC-MS/MS technique allowed Hulla and co-workers to precisely quantify CML in roasted coffee beans, where its contents reached 84.1 mg/kg protein [53]. In a comprehensive ELISA-based study, Vlassara and co-workers compared various foods and beverages in respect of their CML contents [54] and estimated their pro-inflammatory potential *in vivo* [55]. The researchers also found that white heat-treated potatoes contain different amounts of AGEs, depending on the method and duration of heat treatment. Thus, boiling of potatoes for 25 min yielded about 17 kU AGEs/100 g protein, whereas roasting for 1 h resulted in four-fold higher values. Moreover, roasting of potatoes with oil and preparation of french fries’ potato yielded 218 and from 694 to 1522 kU AGEs/100 g protein, respectively [56]. In general, thermally processed plant-derived foods differ essentially in AGE contents, which are usually higher in protein-rich meals with high levels of carbohydrates and/or lipids or fatty acids [57].

As plant-derived thermally processed foods are prone to enhanced accumulation of AGEs, it was assumed that non-heated raw vegetables and fruits might be less dangerous in respect of triggering inflammatory response in mammals. The contents of CML in these foods were comprehensively addressed by Goldberg et al. who showed that abundance of this AGE in fruits varied substantially and accounted 0.01, 0.10 and 0.13 kU/g in bananas, carrots and apples, respectively [54]. Based on ELISA methodology, Vlassara et al. reported essentially higher values which reached 45, 9, 11 and 20 KU CML/100 g protein in apples, tomatoes, bananas and cantaloupe, respectively [56]. In turn, LC-MS/MS analysis of CML adducts revealed high contents of this AGE in plant foods—on average, 26.6 mg/kg protein [53]. This was substantially higher compared to raw foods of animal origin, like raw beef mincemeals, containing only 3.9 mg CML/kg protein [58]. Thus, it can be noted that raw plant-derived foods contain more AGEs (CML) than the foods of animal origin. Hence, one can assume that consumption of plant-derived foods might affect AGE titers in the blood. Indeed, Krajcovicova-Kudlackova et al. demonstrated that blood of vegetarians contained higher amounts of CML in comparison to omnivorous individuals [59]. Potentially, these exogenic AGEs could interact with receptors for AGEs (RAGEs), mentioned in the previous section [39], triggering a systemic inflammatory response [60]. Interestingly, this was not observed when the inflammatory status of vegetarian and omnivorous individuals was compared [59]. This difference in comparison to the effects of thermally-treated foods [54] can probably be attributed to plant metabolites with antiglycative properties. Thus, the question whether exogenous dietary AGEs of plant origin can affect the glycation and/or oxidation status of human tissues, triggering physiological responses therein remains open. To fill this gap, different types of responses need to be addressed and different fractions (protein-bound and unbound) need to be considered.

## 3. Protein Glycation in Plants: From Non-Specificity to Glycation Hotspots

The fact that CML was identified in raw plant-derived foods [54] clearly indicates the presence of AGEs in plant tissues, although their patterns remained unknown for a long time. At the end of the last decade, the patterns of protein-derived glycated adducts (fructosamines and AGEs) were comprehensively characterized by Thornalley’s group [61]. In their work, determination of AGE contents relied on highly-sensitive LC-MS/MS-analysis based on exhaustive enzymatic protein hydrolysis and standard isotope dilution approach, which is currently recognized as a “gold standard” in absolute quantification of protein-bound and unbound glycation adducts [62]. Although this technique does not deliver exact sites of protein modifications (as can be achieved by a bottom-up proteomic approach [63]), it gives an idea on general glycation levels in particular tissues [64,65], providing insight into metabolic alterations causing and/or caused by protein glycation. Having this versatile tool at hand, Thornalley and co-workers demonstrated that glycation adduct patterns followed a circadian rhythm. In other words, the contents of early glycation products (Amadori and Heyns compounds) showed a clear increase upon triggering of photosynthetic carbohydrate production by irradiation, although formation of triose-, tetrose- and pentose-derived early glycation products seems to be more realistic [66]. This dependence of *in vivo* plant glycation rates on sugar equilibrium concentrations raised the question about the relation of glycation phenomenon to environmental factors (high light, drought, high salinity, heating) and associated oxidative stress. Analysis of AGE profiles, dominated by *N^ε^*-(carboxymethyl)lysine, hydroimidazolone modifications of arginyl residues and products of their transformation, pointed out the role of α-dicarbonyls as intermediates of glycation in plants [67]. 

A subsequent in-depth screening of the plant glycated proteome was in agreement with the LC-MS/MS quantitative data and gave good insight into the patterns of specific glycation sites in plant proteins [68]. Indeed, more than half of the 772 AGE-containing proteins, identified in the leaves of *Arabidopsis thaliana* and *Brassica napus*, were modified at arginine residues, and these modifications were mostly represented by GO- and MGO-derived hydroimidazolones (Glagr and MG-H, respectively), as well as the products of their hydrolysis—*N^δ^*-(carboxymethyl)- and -(carboxyethyl)arginine (CMA and CEA). Moreover, the numbers of protein sites modified with GO were higher than those containing MGO-derived modifications. This was in agreement with a high GO/MGO ratio in plant tissues [68]. Surprisingly, despite high contents of such lysine AGEs as *N^ε^*-(carboxymethyl)- and -(carboxyethyl)lysine (CML and CEL), only 21.3% of the total lysyl residue numbers were modified with Amadori and Heyns moieties—early glycation products. This differed dramatically from mammalian systems: for example, the numbers of glycation sites identified in blood plasma proteins [69,70] were one order of magnitude higher than those of AGE-modified ones [3,71,72].

As a next step, we demonstrated that drought contributes to accumulation of AGEs in *Arabidopsis thaliana* leaves [67]. The majority of the 62 drought-specific glycation sites were represented by glyoxal-derived modifications (Glarg, CMA and CML), reflecting a higher increase of GO equilibrium concentrations under stress conditions as compared to other α-dicarbonyls. This up-regulation of dicarbonyl generation can be explained by drought-induced biosynthesis of osmoprotective sugar and sugar-related metabolites (so-called metabolic adjustment) together with increased oxidative stress [66]. Recently, Bechtold and co-workers demonstrated the pro-glycative effect of other environmental stressors such as heat and high light [73]. Flooding stress, which is associated with Reactive Oxygen Species (ROS) production and oxidative burst, results in protein glycation [74]. Salinity was reported as conditions increasing the level of free sugars via several mechanisms [75], probably leading to glycation [76]. Salinity can potentially act on sugars by enhanced HO^•^ production [77] via NaCl-induced depolarization and release of l-ascorbic acid to the apoplastic space, therefore catalyzing Fenton-like reactions of cell wall-bound transition metals [78]. Similar events, i.e., accumulation of reducing sugars and enhancement of ROS production, accompany ageing of plant tissues [79]. Recently, we addressed the question of whether ageing would increase glycation rates in plants, as it was shown for mammals [80]. Surprisingly, although relatively few glycation sites quantitatively responded to ageing, the degree of site-specific glycation at selected amino acid residues was rather high [81]. Thus, similarly to mammalian proteins [82], glycation hotspots, i.e., specific sites of enhanced age-related glycation, could be identified in plant proteins. Interestingly, the existence of such hot spots was recently shown in root nodules of common bean (*Phaseolus vulgaris*), both in plant and rhizobial proteomes [83]. The mechanisms underlying the observed specificity of glycation are currently investigated.

To summarize, glycation of plant proteins has several remarkable features, clearly distinguishing it from the protein Maillard reaction in mammals. First, relatively low numbers of early glycation sites in comparison to AGE-modified residues clearly indicate so-called “oxidative glycosylation” (i.e., oxidative degradation of carbohydrates, formation of α-dicarbonyls with their subsequent interaction with protein side chains) [84,85] as the main glycation pathway in plants. This can most probably be explained by relatively high carbohydrate contents and high oxidative status of plant tissues, i.e., high levels of free radical production [86]. It is important to note that plants contain high amounts of sugar phosphates which have a high glycation potential, however they form relatively low amounts of Amadori of Heyns compounds [68]. Remarkably, the sugar phosphate contents in plants are essentially higher than in mammalian cells [87,88]. Thus, formation of early glycation products might be less characteristic for plants in comparison to mammals. This indicates that glycoxidation, i.e., oxidative degradation of Amadori and Heyns compounds [89] and the Namiki pathway [30], most probably represent only minor mechanisms of AGE formation in plants, whereas so-called oxidative glycosylation, i.e., oxidative degradation of carbohydrates with subsequent interaction of resulted α-dicarbonyls with proteins represents the major pathway [68]. On the other hand, early glycation products, formed in plant tissues, for instance due to interaction with intermediates of the Calvin cycle [90], may be immediately degraded by enzymatic deglycation systems. Such systems are well-known from mammalian biochemistry and might comprise fructosamine-3-phosphate kinase (FN3K) and ribulosamine kinase (FN3K-RP), phosphorylating protein-bound Amadori and Heyns compounds in the third position of the sugar moiety. It results in destabilization of the C–N bond between sugar moiety and the protein and formation of deoxyosones [91,92]. Interestingly, a substantial ribulosamine kinase activity (700-fold exceeding that in mammalian erythrocytes) was observed in spinach leaf extracts [93]. Later, the corresponding enzyme was isolated and its ribulosamine/erythrulosamine 3-kinase activity was characterized [93]. Based on the low levels of pyrraline formation, the same can be concluded for the non-oxidative pathway [94]. And, finally, it appears that glycation hotspots represent one of the most important features of plant glycation [81]. Importantly, there is a clear tendency that these hotspots are characteristic for enzymes and regulatory proteins, raising that the intriguing question whether glycation may be fulfilling a possible regulatory role in plants, probably becoming relatively more important during periods of enhanced sugar and oxidative stress.

## 4. Possible Role of Protein Glycation in Plant Physiology

In contrast to mammals, the role of glycation in plants is still poorly understood. Similar to ROS acting both as a detrimental factor and as a beneficial signal, depending on their concentration and duration of exposure [95], two main functional aspects can be proposed for protein glycation and glycoxidation as well. On one hand, excessive protein glycation results in protein damage and directs polypeptides for degradation by the proteasome. On the other hand, limited, and to some extent, more controlled glycation may be involved in signalling and regulatory events in plants.

### 4.1. Glycation as the Marker of Ageing, Senescence and Tag for Protein Degradation

It was shown *in vitro* that glycation of ribulose-1,5-bisphosphate carboxylase/oxygenase (RuBisCO), the most crucial enzyme of photosynthesis and probably the most abundant enzyme on the planet, with ascorbic acid resulted in activity loss and increased susceptibility to proteosomal degradation [96]. Probably, more extensive glycoxidative modifications of RuBisCO impact on disintegration of chloroplasts, as occurring during senescence and nitrogen starvation [97]. It can be reasonably assumed that glycation represents one of the marker non-enzymatic post-translational modifications, directing plant proteins to degradation and corresponding organs to senescence and death. Most probably, the process is accompanied with oxidation [25] and modification with reactive carbonyls—products of lipid peroxidation [98]. The combination of these post-translational modifications (PTMs) is often referred to as “carbonylation” and recognized as a marker for plant senescence and a tag for proteolysis [99]. In mature seeds, glycation represents one of the major markers of seed ageing, in concert with the decrease of their quality and longevity during natural and accelerated storage [100,101].

It is well known that in mammals, depending on its severity (that is the matter of duration and concentration of carbonyl glycation agents), glycation can either enhance or inhibit protein degradation [39]. Thereby, glycation can interfere both with proteasomal [102] and lysosomal/autophagy systems [103]. Moreover, the function of mammalian proteasomes can be modulated by MGO, a potent glycation agent, both *in vitro* and *in vivo* [104]. This α-dicarbonyl was shown to suppress chemotrypsin-like proteasome activity via modification of the 20S subunit with MGO. This suppression might result in accumulation of glycated proteins in cells, causing aggregation [105]. Such aggregates cannot be efficiently degraded by the proteasome [106]. One can assume that similar mechanisms might exist in plants and function as a fine tuning mechanism of protein degradation rates. However, so far, no information on the impact of glycation on protein degradation and aggregation is available for plants.

As mentioned above, based on our own observations [67] and the data obtained by other groups [61,107], the main glycation pathways in plants are mediated by α-dicarbonyls. Although the most abundant plant α-dicarbonyl is GO, MGO is much better studied in terms of its formation, reactivity and biological role in plants. This metabolite ultimately forms during glycolysis [108], thus it is ubiquitous in plants, animals, yeast and bacteria, accounting 0.1–0.4 % of its yield [109]. MGO (both externally applied and stress-induced) results in growth retardation in experimental plants [110,111]. However, on the other hand, one should keep in mind that under altering environmental conditions, MGO may also serve as a messenger of adaptation signalling pathways [112].

### 4.2. Glycation as a Possible Mechanism behind MGO Signalling

The pathways of MGO formation rely on enzymatic (cytochrome P450, monooxygenase and methylglyoxal synthase) and non-enzymatic (autoxidation of sugars, ketone bodies and lipid peroxidation) mechanisms [113,114]. Currently, non-enzymatic conversion of triose phosphate isomers—dihydroxyacetone phosphate (DHAP) and glyceraldehyde-3-phosphate (GAP), is considered to be the main route of MGO formation in plants (Figure 3) [115]. Although the constitutive cellular MGO contents are rather low, they might rise significantly under environmental stress [116,117]. Accompanying oxidative stress results in depletion of glutathione pool, suppression of MGO detoxification [118,119] and enhancement of metal-catalyzed autoxidation of sugars and lipids due to increased production of hydrogen peroxide, as well as sugar and lipid hydroperoxides [120]. Under the conditions of oxidative stress, activities of glycolytic enzymes and specifically glyceraldehyde phosphate dehydrogenase (GPDH) are impaired [121]. It leads to accumulation of glyceraldehydes-3-phosphate/dihydroxyacetone phosphate, non-enzymatic formation of MGO along the path described above [115].

The generated MGO may be involved in triggering adaptive reactions, important for plant survival under environmental stress (Figure 3) [112]. In their experiments relying on GeneChip microchip technology, Kaur et al. investigated MGO-induced changes in global rice gene expression profiles [112]. The authors reported a 1.5-fold increase in gene expression of signalling protein kinases (seven of which represented the MAP-signalling pathway) and proteins involved in stress adaptation. Thereby, a specific MGO-responsive element (MGRE) was identified [112]. Recently, Hoque and co-workers showed that increase of MGO contents in *Arabidopsis thaliana* leaves affected the expression of the genes RD29B and RAB18 involved in ABA-mediated stress responses and stomatal closure [122,123].

As the half-life of MGO is rather short, its concentration is higher at the sites of its formation [124] which are usually localized in organelles with increased ROS production, such as chloroplasts and mitochondria [86,125]. In line with this, Takagi et al. showed that MGO contents increased in isolated chloroplasts with prolonged light exposure times and intensity [98]. As excessive MGO is extremely toxic and can promote cell death [126], plants have strong protective mechanisms of MGO detoxification. Thus, Atlante et al. described targeted transformation of cytoplasmic MGO in *d*-lactate by MGO-reductase in *Helianthus* cells [127]. However, the major player in dicarbonyl detoxification (not only MGO, however also GO) is glyoxalase system, comprising two enzymes—glyoxalases I and II (Figure 3) [128]. At the first step of this detoxification procedure, α-dicarbonyl (GO or MGO) interacts with a glutathione (GSH) molecule, yielding corresponding hemithioacetal (HTA), which is afterwards metabolized by glyoxalase I to *S*-2-hydroxyacylglutathione [129]. This intermediate is further cleaved by glyoxalase II to GSH and non-toxic *d*-lactate [130]. An alternative enzyme, glyoxalase III [131], found in approximately 100 monocotyledonous and dicotyledonous species, can metabolize MGO to *d*-lactate without involvement of GSH [131].

At the molecular level, the mechanisms behind MGO signalling are still poorly understood. However, the existence of a finely tuned system of MGO generation and detoxification strongly suggests that AGE formation plays a role in signal transduction and/or effector mechanisms. Hypothetically, MGO can be involved in ROS- and K^+^-dependent metabolic adjustment and inducible autophagy shifting plant metabolism from anabolic to catabolic pathways during stress responses [132].

### 4.3. Glycation and Non-Enzymatic Antiglycative/Antioxidant Defense

GSH is not only involved in detoxification of MGO, however it also impacts on prevention of its overproduction via quenching ROS and suppressing oxidative stress [133], which can be triggered by various forms of environmental stress [79,95,134]. Indeed, it was shown that drought conditions and salt stress result in substantial enhancement of protein glycation in plant tissues [67]. To a large extent, this effect can be mediated by enhanced generation of α-dicarbonyls during monosaccharide and lipid autoxidation [27]. Involvement of enzymatic and non-enzymatic antioxidant defense leads to neutralization of free radicals and hydrogen peroxide and might result in a decrease of carbonyl production, often referred to as carbonyl stress [135]. The enzymatic antioxidant machinery was intensively reviewed by Gill and Tuteja [136]. The cellular non-enzymatic antioxidants are represented by l-ascorbic acid, GSH, carotenoids, flavonoids and tocopherol [137]. In contrast to antioxidant enzymes [138,139], these molecules are not specific for individual ROS, i.e., they are not capable of neutralizing all types of free radicals. Importantly, glutathione can be transported by enzyme transpeptidase across the cell membrane that allows maintaining redox status of the whole tissue/organ [140]. Thereby, reduced (GSH) and oxidized (GSSG) forms of glutathione compete for binding with this enzyme which has much higher specificity for the conjugates of GSH in comparison to those of GSSG [141]. Such enzymes as glutathione peroxidase, glutathione transferase [142,143] and glutathione reductase are involved in glutathione homeostasis [144]. Other antioxidants, such as ascorbate [145] and dihydrolipoate [146], also contribute in GSH/GSSG equilibrium in cells. On the other hand, tocopherols [147], carotenoids [148] and flavonoids [149] might act as free radical traps, maintaining redox homeostasis by direct binding to ROS [147].

It is important to note, however, that some antioxidants may exert pro-oxidant effects in the presence of transition metals [150] due to enhanced generation of hydroxyl radical in Fenton reaction [95,151]. Thus, such compounds as l-ascorbic acid [152,153] can neutralize ROS, however in the presence of transition metals, they might result in oxidative damage to lipids, proteins and nucleic acids, triggering oxidative stress signalling events, growth retardation and redox-dependent programmed cell death [78,154]. Contrary to animals, ascorbate levels are extremely high within plant tissues, reaching tens or hundreds of millimoles in most cell compartments (particularly in ROS-enriched chloroplasts and mitochondria) and redistributing by efficient active and passive transport mechanisms [78]. Products of l-ascorbate oxidative degradation, such as l-threose, are very active glycation agents [155]. Ascorbate-induced glycation can have significant consequences for a cell, for example, glycation by l-ascorbic acid induced loss of activity of ribulose-1,5-bisphosphate carboxylase/oxygenase and increased its susceptibility to proteases in *Cucumis sativus* L [156]. A special and very interesting case of glycation by l-ascorbic acid is ascobylation [157]. Nevertheless, a full potential of this phenomenon has never been explored in plants.

### 4.4. Possible Interplay between Glycation and Sugar Signalling

Historically, our views on the control of plant growth, development and stress responses have gradually changed. Three decades ago, plant hormones were considered the only signalling entities controlling all aspects in a plant’s life cycle. At the end of the nineties, sugars came into the picture as no longer only serving as substrates for energy provision, however also acting as real signalling entities with hormone-like actions [158]. Since then, our awareness on the relative importance of sugars during plant developmental processes, and especially during stress responses, has only been growing [159]. At present, a picture is emerging in which sugars may be considered as “master regulators” [160], controlling plant hormone synthesis (e.g., auxins) [161,162] and overruling the dominance of plant hormonal regulation during plant physiological processes (e.g., apical dominance) [163].

The three major neutral small soluble sugars recognized to play a role in plant sugar signalling are sucrose (Suc, non-reducing) and its reducing hexose constituents, glucose (Glc) and fructose (Fru) [164]. Evidently, Suc splitting enzymes (sucrose synthase, different types of invertases) can drastically change the Suc/hexose ratio and sugar signalling outcomes. The hexoses stand out from most reactive molecules from the perspective of glycation, with Fru being more reactive than Glc [68]. Thus, like is the case for all living creatures, plants developed mechanisms to strictly control Fru concentrations in their cytosolic environment where the mainstream metabolic processes occur. Fructokinase is a major player in this context [165], transforming Fru into fructose-6-phosphate (Fru6P) in metabolic equilibrium with other sugar phosphates such as glucose-6-phosphate (Glc6P), glucose-1-phosphate (Glc1P) and UDP-glucose (UDPGlc). Sequestration of Fru in the plant vacuole may be another option, although this raises the question of how plants can avoid extensive glycation and inhibition of vacuolar enzymes. UDPGlc can either combine with Glc6P to form trehalose-6-phosphate (T6P; trehalose-6-phosphate synthase (TPS) activity) and trehalose (trehalose-6-phosphate phosphatase (TPP) activity) or with Fru6P to form sucrose-6-phosphate (sucrose-6-phosphate synthase (SPS) activity) and sucrose (sucrose-6-phosphate phosphatase (SPP) activity) [166]. In view of glycation, it is well known that Glc6P stands out as most reactive species, being even more reactive than Fru6P, Fru and Glc [68]. The reactivity of T6P and UDPGlc are not known in this context and deserve further exploration.

Historically, research on Glc signalling through hexokinase (AtHXK1) came first [158]. Focusing on the context of very young seedlings depending on seed reserves, it was demonstrated that this kinase is a moonlighting enzyme with dual function: on the one hand synthesizing G6P to fuel glycolysis; on the other hand functioning as a Glc sensor, mediating signalling and differential gene expression. However, the signalling role of AtHXK1 seems to be fading away during further plant development [167]. Fru signalling has been suggested twice [168,169], however no Fru sensor or clear set of target genes could be defined and, therefore, research efforts ceased in this direction. By contrast, Suc signalling is widely recognized [170,171], affecting the differential expression of hundreds of genes, with confirmed importance during the whole life cycle of the plant and becoming increasingly important during stress responses. Interestingly, the synthesis of antimicrobial compounds belongning to the phenylpropanoid branch depends mainly on Suc siugnalling and not Glc signaling [172]. This suggests that Suc-mediated signaling processes may stimulate plant defense responses, leading to enhanced biosynthesis of secondary metabolites of the phenylpropanoid branch such as anthocyanins [172,173,174,175].

Despite extensive efforts to find one or more Suc receptors, no such entities could be found, however the hunt keeps on going. During the last decade and especially during the last five years, focus in the plant sugar signalling field shifted towards the balanced control of two major kinases over plant development and stress responses: SnRK1 [176] and SnRK2 [177] on the one hand (stimulating stress responses) and TOR kinase on the other hand (stimulating growth) [178]. Their counterparts in the animal/human context (AMPK, TOR) are deeply studied and their activities seem to be greatly changing depending on the sugar concentration and ROS context, intrinsically connected to autophagy [103,179]. The generally accepted view is that SnRK1/AMPK activities are central during nutrient (sugar) starvation processes, while TOR kinase takes over control when nutrients (sugars) are readily available [164]. However, this view has been recently challenged by pointing to the emerging evidence that plant SnRKs (especially SnRK2) may actually be put “on” again under conditions of sugar excess, however this requires further experimental verification [103]. Contrary to the animal context where AMPK activity is directly controlled by the AMP/ATP ratio, this is not the case for SnRK1 [164]. T6P (next to Glc6P) is emerging as the main signal reflecting Suc availability [180] and inhibiting SnRK1 activity and focus on defense, paving the way for TOR kinase control by yet another hexokinase independent way of Glc signalling (Figure 4). Major challenges remain, however, both in the SnRK1 and TOR kinase contexts. Moreover, UDP-Glc is emerging as a potentially novel signalling molecule, in line with its signalling capability in animals [181]. Additionally, the SnRK1/TOR pathway can potentially act downstream of both Ca^2+^ and ROS signalling [132].

Concerning the T6P/SnRK1 context, the direct interaction between T6P and KIN10, the catalytic unit of SnRK1, has been demonstrated once by using one *in vitro* technology [182]. However, the exact interaction locus on KIN10 remains undetermined and further research into the *in vivo* context with genetic tools and ligand co-crystallization and/or sugar docking in independent labs is warranted to further confirm this interaction. Unfortunately, the interaction between KIN10 and Glc6P, Glc1P and T6P have not yet been fully compared to each other with the abovementioned *in vitro* technology, although all are recognized as important negative regulators of SnRK1 activity [183]. In this respect, bearing in mind the enormous reactivity of G6P in the context of glycation, it is not unthinkable that there could be room for glycation inhibiting SnRK1 activity.

In the Glc/TOR regulatory context, it was proven that Glc needs to enter glycolysis first (through hexokinase, mainly HXK1) to establish the signaling effect [184] (Figure 4). At the same time, however, it was demonstrated that this signalling occurs independently from HXK1. Thus, it was suggested that a more downstream but yet unidentified glycolytic intermediate may directly regulate TOR kinase activity. Intriguingly, during “sugar feast” in animals (e.g., diabetes), leading to an excess of reducing sugars, glycation clearly comes into the picture as a prominent path. Under such circumstances, the glycolytic intermediate 1,3-bisphosphoglycerate reacts with lysine residues present in or around the active sites of glycolytic enzymes. This redirects carbon to alternative pathways [185], presumably containing enzymes that should be less sensitive to glycation. Also, as was mentioned above, glyceraldehyde-3-phosphate can be non-enzymatically converted to MGO—a potent glycation agent [68,115].

Overall, this clearly suggests once again that evolution played around with glycation sensitivity, with some enzymes or transcription factors becoming particularly sensitive to inhibition by glycation, while others most probably becoming more resistant. Likely, such evolutionary processes also occurred in plants, which are even more prone to glycation effects than animals. Therefore, it is even more likely that plants fully integrated glycation effects into their overall regulatory processes. In this regard, it would be particularly interesting to investigate further how increased ROS and sugar, promoting glycation, control the delicate SnRK1/TOR balance, regulating the crucial trade-off between growth and defense in plants. Hypothetically, this links to the ROS–Ca^2+^ signalling machinery involving massive loss of K^+^ under stress [132,186]. Needless to say, in this context, it would be necessary to consider glycation effects on sucrose metabolizing enzymes as well (Figure 4) since they are required to synthesize the T6P, GlcP and Glc signalling entities involved in SnRK1 and TOR regulation. The comparative glycation potential of these sugars and sugar phosphates in this regulatory context needs to be carefully considered. Importantly, there is an urge to also consider the glycation potential of UDP-Glc from this point since nucleotides might be potent glycation agents and signalling molecules as well [181,187].

## 5. Conclusions

Protein glycation, often referred to as Maillard reaction of proteins, is a well-characterized phenomenon, accompanying pathogenesis of human metabolic diseases and thermal processing of foods. The activities of resulted protein Maillard reaction products (Amadori, Heyns compounds and AGEs) in mammalian systems (cell cultures, experimental animals, human volunteers) are partly characterized, although the opinions on biological interpretation of the observed effects differ essentially within Maillard reaction society. Surprisingly, plant proteins are essentially more heavily glycated, hinting at an important role of the glycation and glycoxidation phenomenon in plant physiology. Indeed, on the one hand, AGEs can serve as protein degradation tags, impacting on regulatory pathways at the level of protein degradation. On the other hand, glycation and glycoxidation may interplay with sugar metabolism and signalling in plants. A better understanding of this crosstalk may be key to understanding the physiological role of glycation and glycoxidation in plants.

## Figures and Tables

**Figure 1 ijms-20-02366-f001:**
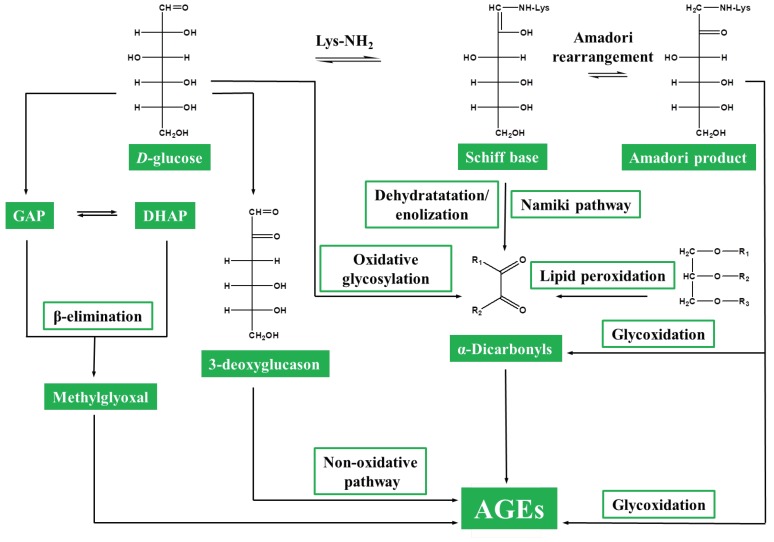
Pathways of AGE formation. AGEs, advanced glycation end products; GAP, glyceraldehyde-3-phosphate; DHAP, dihydroxyacetone phosphate.

**Figure 2 ijms-20-02366-f002:**
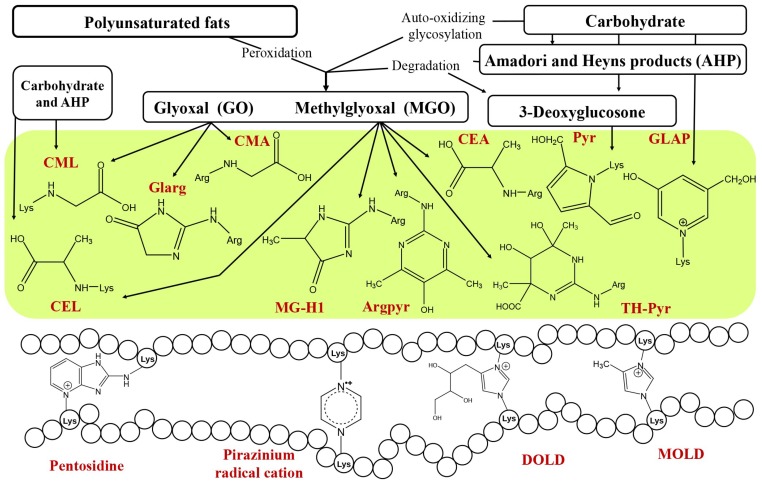
Advanced glycation end products (AGEs), their precursors and intermediates in the context of underlying formation mechanisms. Abbreviations: CML, *N^ε^*-(carboxymethyl)-lysine; CEL, *N^ε^*-(carboxyethyl)-lysine; CEA, *N^δ^*-(carboxyethyl)-arginine; CMA, *N^δ^*-(carboxymethyl)-arginine; Glarg, glyoxal-derived hydroimidazolone; MG-H1, methylglyoxal-derived hydroimidazolone (*N*^δ^-(5-methyl-4-oxo-5-hydroimidazolinone-2-yl)-l-ornithine); Argpyr, argpyrimidine; TH-Pyr, tetrahydropyrimidine; Pyr, pyrraline; GLAP, glyceraldehyde derived pyridinium compound; MOLD, lysine-lysine imidazolium cross-link, derived from methylglyoxal; DOLD, lysine-lysine imidazolium cross-link derived from 3-deoxyglucosone.

**Figure 3 ijms-20-02366-f003:**
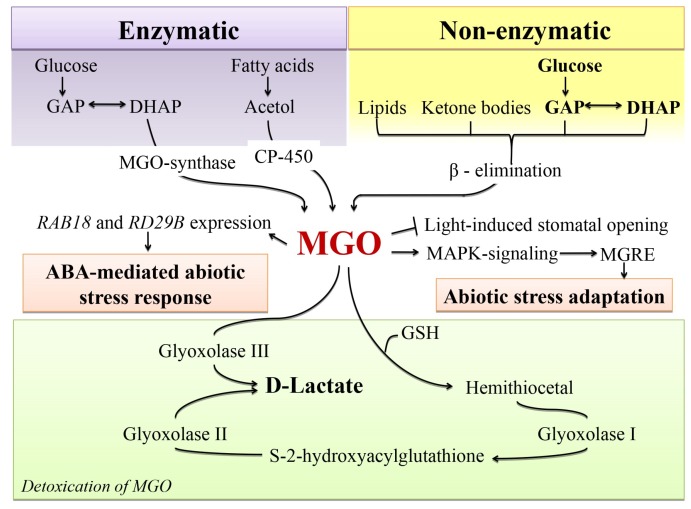
The pathways of MGO formation, detoxification and signalling in plant cells. MGO, methylglyoxal; GAP, glyceraldehyde-3-phosphate; DHAP, dihydroxyacetone phosphate; ABA, abscisic acid; MGRE, methylglyoxal responsive element; GSH, glutathione. Arrows and T-bars represent stimulatory and inhibitory effects, respectively.

**Figure 4 ijms-20-02366-f004:**
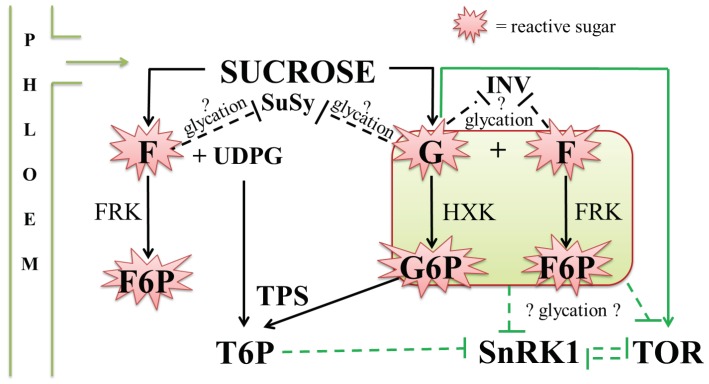
Hypothetical model for interplay between glycation and sugar metabolism and signalling in plants. F, fructose; G, glucose; F6P, fructose-6-phosphate; G6P, glucose-6-phosphate; SuSy, sucrose synthase; INV, invertases; T6P, trehalose-6-phosphate; SnRK1, SNF1-related protein kinase 1; TOR, target of rapamycin kinase; UDPG, uridine diphosphate glucose; HXK, hexokinase; FRK, fructokinase; TPS, trehalose-6-phosphate synthase. Green arrows represent sugar signaling pathways and black arrows represent sugar metabolism. Arrows and T-bars represent stimulatory and inhibitory effects, respectively.

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
