# Peer review of "Glycation of Plant Proteins: Regulatory Roles and Interplay with Sugar Signalling?"

_ijms, 2019, doi:10.3390/ijms20092366_

Reviewer 1 Report

Manuscript  (ID: ijms-495545) entitled "Glycation of plant proteins: regulatory roles and interplay with sugar signaling?" intended for  publication in International Journal of Molecular Sciences is an interesting paper, related to the IJMS. I think the manuscript is a well written review that includes all important and new information on plant protein glycation. However, there are small mistakes in the text, thus manuscript is not suitable for publication in the present form and needs some improvements.

Minor remarks:

a) references list must be carefully checked and improved - too many small mistakes, e.g. see capital letters, not necessary in many bibliographical references

b) the Authors should consider some small improvements in the figures presentation and description, e.g. explain all shortcuts to the Figures 1-3

c) some parts of manuscript could be shorter, e.g.  introduction…  

Author Response

We thank the reviewer for the thoughtful review and highly appreciate the valuable comments and suggestions to improve the manuscript. Following these advices we performed all required changes in corresponding sections, as indicated in the following rebuttal addressing each aspect. We apologize, that because of parallel preparation a grant application we missed two colleagues in the author list – Alexander Tsarev and Henry Janse van Rensburg. Here we put them into the author list and addressed their impact in the “Author contributions” section.

Reviewer: 1

Minor remarks:

Minor remark 1: References list must be carefully checked and improved - too many small mistakes, e.g. see capital letters, not necessary in many bibliographical references

Answer: We checked and corrected the reference list where was necessary.

Minor remark 2: The authors should consider some small improvements in the figures presentation and description, e.g. explain all shortcuts to the Figures 1-3”

Answer: We agree with the reviewer - all shortcuts, corresponding to the Figures 1-3 are explained (lines 75, 102, 338).

Minor remark 3: Some parts of manuscript could be shorter, e.g.  introduction…”

Answer: We understand this point, highlighted by the Reviewer, and reduced the introduction part slightly. But we think, as the topic is situated at the cross-road of several chemical and biological areas (and hence, the paper can be read by scientists with dramatically different expertise), intensive reduction of the introduction size might compromise clearness of presentation. We hope, that our presentation of protein Maillard reaction is, from one hand, understandable, from another – brief enough.

Reviewer 2 Report

The review entitled "Glycation of plant proteins: regulatory roles and interplay with sugar signaling?" adresses an interesting and emerging topic in plant biology and paves the way, through many examples, for further investigating the interest and the prevailance  of this process in the regulatory network related to many physiological plant processes.
This paper is well written and concise and is easy to read.
I only have some minors comments:
1) I will invite the authors to illustrate, through a schema or figure, the two kinds of glycation existing in plants, including glycoxydation.
2) I am a little bit confused about the figure 4.  It will be interesting that the authors specify which part corresponding to sugar metabolism and and which one is related to sugars signaling
3) I am wondering if there are glycation or glycoxydation motifs  that allow the authors to propose an hypothetical glycation of SuSy, Invertase et SnRK1.
4) The authorscould add recent reviews linked to the master role of sugars in plant physiology such as
    a) Wingler , 2018 Plant Physiol. 2018 Feb;176(2):1075-1084
    b) Sakr et al., 2018 Int J Mol Sci. 2018 Aug 24;19(9)

Author Response

We thank the reviewer for the thoughtful review and highly appreciate the valuable comments and suggestions to improve the manuscript. Following these advices we performed all required changes in corresponding sections, as indicated in the following rebuttal addressing each aspect.

Reviewer: 2

Minor remarks:

Minor remark 1: I will invite the authors to illustrate, through a schema or figure, the two kinds of glycation existing in plants, including glycoxydation

Answer: We agree with the reviewer. Accordingly, two main glycation pathways – glycoxidation and oxidative glycosylation are depicted in Figure 1.

Minor remark 2: I am a little bit confused about the figure 4.  It will be interesting that the authors specify which part corresponding to sugar metabolism and which one is related to sugars signaling

Answer We agree with the reviewer – this would be advantageous. We changed Figure 4 so that glycation can potentially affect both SnRK1 and TOR kinases. Also, the arrow colors have been changed to distinguish between sugar signaling (green) and metabolism (black) as requested.

Minor remark 3: I am wondering if there are glycation or glycoxydation motifs that allow the authors to propose an hypothetical glycation of SuSy, Invertase et SnRK1

Answer: A “real” motif prediction for glycation/glycoxidation is not yet possible.  Even when having perfect 3D structures at hand, everything remains very hypothetical. Thus, likely one overestimates the number of potential glycation/glycoxidation sites. No pdb files are available for plant SnRK1 and TOR complexes (heterotrimers) and the quality of obtained models is highly questionable. However, the 3D structure of human AMPK is available. There, a clear glycation hot spot occurs at the AMP allosteric site (Gugliucci, 2016), attracting our attention towards this matter. However, all of this remains to be demonstrated (i.e. experimentally confirmed), and therefore we refrain from going to deep in highly speculative suggestions. The glycation/glycoxidation of the energy sensing network just needs to be experimentally investigated.

Minor remark 4: The authors could add recent reviews linked to the master role of sugars in plant physiology such as: Wingler , 2018 Plant Physiol. 2018 Feb;176(2):1075-1084; Sakr et al., 2018 Int J Mol Sci. 2018 Aug 24;19(9)

Answer: We added recent reviews: [162] - Sakr et al., 2018 Int J Mol Sci. 2018 Aug 24;19(9); [171] - Wingler , 2018 Plant Physiol. 2018 Feb;176(2):1075-1084 (lines 385, 412).

Reviewer 3 Report

The review article entitled “Glycation of plant proteins: regulatory roles and interplay with sugar signaling?” by Dr. Julia Shumilina are well written and the authors mention glycation of plant proteins and plant-derived foods. I do not have any major comment though, I would like to ask only few things.

1. The authors used 2 types of abbreviation such as MG and MGO for methylglyoxal. Does the authors have any specific reason to distinguish those 2 words. If not please choose one of them.

2. In figure 3, no Glyoxalase I is existed. I guess right lower “Glyoxalase III” is “Glyoxalase I”. Please confirm it.

Author Response

We thank the reviewer for the thoughtful review and highly appreciate the valuable comments and suggestions to improve the manuscript. Following these advices we performed all required changes in corresponding sections, as indicated in the following rebuttal addressing each aspect. We apologize, that because of parallel preparation a grant application we missed two colleagues in the author list – Alexander Tsarev and Henry Janse van Rensburg. Here we put them into the author list and addressed their impact in the “Author contributions” section.

Reviewer: 3

Minor remarks:

Minor remark 1: The authors used 2 types of abbreviation such as MG and MGO for methylglyoxal. Do the authors have any specific reason to distinguish those 2 words. If not please choose one of them

Answer: We apologize for this inconsistence. Accordingly, MG is replaced with MGO throughout the text (lines 311, 314).

Minor remark 2: In figure 3, no Glyoxalase I is existed. I guess right lower “Glyoxalase III” is “Glyoxalase I”. Please confirm it”

Answer: Yes, we apologize because of this. The Figure 3 is corrected accordingly.